# MLE-UVAD: Minimal Latent Entropy Autoencoder for Fully Unsupervised Video Anomaly Detection

## Abstract

In this paper, we address the challenging problem of *single-scene, fully unsupervised* video anomaly detection (VAD), where raw videos containing both normal and abnormal events are used directly for training and testing without any labels. This differs sharply from prior work that either requires extensive labeling (fully or weakly supervised) or depends on normal-only videos (one-class classification), which are vulnerable to distribution shifts and contamination. We propose an entropy-guided autoencoder that detects anomalies through reconstruction error by reconstructing normal frames well while making anomalies reconstruct poorly. The key idea is to combine the standard reconstruction loss with a novel Minimal Latent Entropy (MLE) loss in the autoencoder. Reconstruction loss alone maps normal and abnormal inputs to distinct latent clusters due to their inherent differences, but also risks reconstructing anomalies too well to detect. Therefore, MLE loss addresses this by minimizing the entropy of latent embeddings, encouraging them to concentrate around high-density regions. Since normal frames dominate the raw video, sparse anomalous embeddings are pulled into the normal cluster, so the decoder emphasizes normal patterns and produces poor reconstructions for anomalies. This dual-loss design produces a clear reconstruction gap that enables effective anomaly detection. Extensive experiments on two widely used benchmarks and a challenging self-collected driving dataset demonstrate that our method achieves robust and superior performance over baselines.

## 1 Introduction

Video anomaly detection (VAD) aims to identify abnormal events that deviate from regular patterns, such as violence, accidents, and other unexpected occurrences. Accurate detection is crucial for timely response in safety-critical applications, including surveillance, autonomous driving, and unmanned aerial vehicles (Liu et al., 2025b).

Existing VAD approaches can be divided into three main categories (Liu et al., 2025a): fully supervised (Pang et al., 2020), weakly supervised (Karim et al., 2024), and unsupervised/one-class classification (OCC) (Kommanduri & Ghorai, 2024). However, most existing methods rely on extensive human annotation. Fully supervised approaches demand frame-level labels for every training video, which is prohibitively labor-intensive. Weakly supervised methods ease this burden by using video-level labels—marking a video as abnormal if any frame contains an anomaly. Yet, such videos often include normal segments, making it difficult to disentangle abnormal frames from normal ones (Tian et al., 2021). A third line of work, unsupervised or one-class classification (OCC), trains exclusively on normal videos and requires no abnormal labels (Zong et al., 2018; Wang et al., 2021). While this further reduces labeling costs, two major limitations remain in OCC:

**Limitation 1: Sensitivity to Contamination.** The effectiveness of OCC models depends on training exclusively with clean, anomaly-free data. Even a small fraction of anomalies in the training set can drastically degrade the detection performance (Ramachandra et al., 2022; Yu et al., 2019). Therefore, compared to an unsupervised setting, OCC models require significant investment into a fully cleaned dataset, obtaining which is non-trivial and labor-intensive (Al-lahham et al., 2024).

**Limitation 2: High Cost for Labeling Normal Data.** The OCC setting has no anomaly labels, but selecting "normal" videos is a subjective human task, the cost of which can vary across annotators (Ramachandra et al., 2022). Moreover, OCC methods assume that the distribution of normal videos remains consistent between training and testing (Dawoud et al., 2025). However, even minor shifts in camera characteristics, lighting conditions, or location can cause a distribution shift. When training data distribution shifts, people need to relabel the normal-only videos from scratch.

To address these limitations and avoid any ambiguity of the supervision, we propose a *fully unsupervised setting*, in which raw videos containing both normal and abnormal frames are used for training and testing without any labels, such as surveillance footage without manual annotation or privacy-sensitive labeling. Specifically, we focus on single-scene video anomaly detection where a fixed camera monitors a specific environment. Most prior work overlooks the key distinction between single-scene and multi-scene settings (Ramachandra et al., 2022). Multi-scene training uses videos from different environments and generalizes all scenes to what is normal vs. anomalous (Zhang et al., 2024). However, this contradicts the fact that anomalies are location-dependent (Zhu et al., 2024): the same behavior may be normal in one region but abnormal in another (e.g., walking in a permitted area vs. a restricted zone). In contrast, the single-scene setting focuses on the spatial or temporal regularities to learn location-dependent rules, yielding clearer decision boundaries and immediate practical use (Verma et al., 2025).

Previous OCC methods often train autoencoders on normal videos with reconstruction loss, where higher reconstruction error flags out-of-distribution anomalies (Kommanduri & Ghorai, 2024). Inspired by this, we address our fully unsupervised VAD problem with *MLE-UVAD*, an entropy-guided autoencoder that augments the standard reconstruction loss with a minimal latent entropy (MLE) loss. With reconstruction loss alone in a fully unsupervised setting (e.g., unlabeled video), autoencoders often reconstruct both normal and anomalous frames accurately. Therefore, no clear threshold on the reconstruction error can be set to detect anomalies. Next, the MLE was designed to maintain normal reconstructions while flagging anomalies by minimizing the latent entropy.

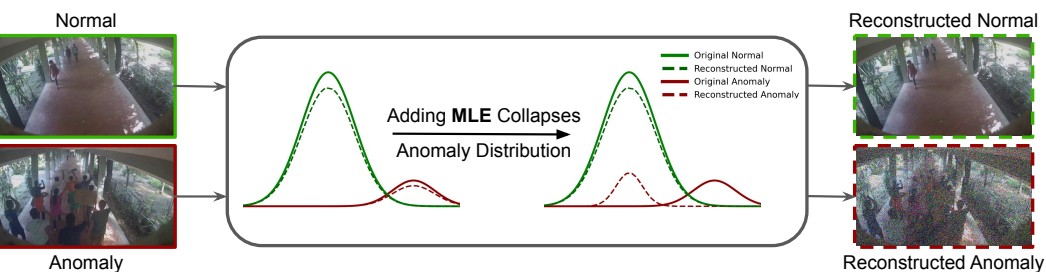

Figure 1: Collapse of the anomalous latent distribution. Green: normal frames. Red: abnormal frames. Solid lines: latent distributions of the original inputs. Dashed lines: latent distributions of the reconstructions. The key idea is to pull sparse anomalous embeddings toward the dense normal embeddings, so normal frames remain well reconstructed while anomalies reconstruct poorly.

Our key novelty is that we collapse anomalous latent representations toward the normal cluster with our proposed MLE loss, as shown in Fig. 1. Mechanistically, reconstruction loss alone always maps normal and abnormal inputs to distinct latent distributions for accurate reconstruction. The MLE loss regularizes these latent distributions by minimizing latent entropy. Because normal videos dominate the unlabeled dataset, entropy minimization concentrates probability mass around the normal clusters and squeezes the sparse, shifted anomalous embeddings toward normal ones. Therefore, anomalous frames are reconstructed as *normal-looking* outputs and accordingly incur larger errors when compared to their original abnormal inputs. Consequently, normal videos remain accurately reconstructed, whereas anomalous videos are poorly reconstructed. This reconstruction gap provides a clear and practical threshold for anomaly detection.

**Contributions.** The contributions of this paper are (1) A single-scene, fully unsupervised VAD setting that eliminates the need for human labeling; (2) An entropy-guided autoencoder with an MLE loss for fully unsupervised VAD; (3) An experimental validation on two public benchmarks and an autonomous racing dataset, consistently outperforming the baselines.

## 2 BACKGROUND AND RELATED WORK

**Weakly Supervised Video Anomaly Detection (WS-VAD).** WS-VAD assumes access to video-level labels: each training video is annotated as either normal or abnormal, but the time points of anomalies are unknown. This problem is typically formulated in a multiple-instance learning (MIL) framework (Sultani et al., 2018), where a video is treated as a bag of snippets, and the model is trained to assign high anomaly scores to at least one snippet in abnormal bags. Subsequent works improved upon this paradigm by introducing contrastive objectives (Tian et al., 2021), noise-robust label cleaning (Zhong et al., 2019), and discriminative representation learning (Wan et al., 2021). However, WS-VAD still requires human annotation at the video level. Moreover, abnormal-labeled videos often contain many normal snippets, resulting in significant label noise and difficulty in precisely localizing the anomalies.

**Unsupervised Video Anomaly Detection (UVAD).** Most prior UVAD uses one-class classification: models train on normal-only data and flag deviations as anomalies (Wang et al., 2021). They fall into three categories: reconstruction-based (Zhou & Paffenroth, 2017), prediction-based (Sun et al., 2024), and clustering-based (Zong et al., 2018). These methods are highly sensitive to the previously mentioned limitations: contaminated training data and distribution shift.

Fully unsupervised anomaly detection is quite sparse in the literature. Generative Cooperative Learning (GCL) (Zaheer et al., 2022) trains an autoencoder and a discriminator cooperatively on unlabeled videos, where pseudo-labels are iteratively refined to discourage anomaly reconstruction. While conceptually appealing, GCL can be unstable across scenarios due to a lack of clear pseudo-labeling rules, which accumulates errors to destabilize training. Temporal Masked AutoEncoding (TMAE) (Hu et al., 2022) detects anomalies by training a Vision Transformer (ViT) to predict masked patches in spatiotemporal cubes of unlabeled video, where rare events yield higher prediction errors. However, its reliance on ViT requires substantial computational overhead, hindering real-time deployment in our single-scene setting. Both are compared with our method in the results.

**Problem Formulation.** Given the raw *fully unsupervised video* directly from a fixed camera, our task is to detect whether an observation/frame $x_i$ is an anomaly. Given the video

$$V = V^{\text{normal}} \cup V^{\text{abnormal}} = \{x_t\}_{t=1}^T, \qquad x_t \in \mathbb{R}^{C \times H \times W}, \tag{1}$$

where each observation $x_t$ is an image with $C$ channels and spatial resolution $H \times W$. This problem requires calculating the anomaly score $S(x_t \mid V)$ given the whole video $V$ — and then comparing it with a tunable threshold $\tau$. If the $S(x_t \mid V) > \tau$, then $x_t$ is flagged as an anomaly.

## 3 METHODOLOGY

This section first overviews the MLE-UVAD pipeline and formalizes our proposed MLE loss. Next, using t-SNE of the latent space, we show why it strengthens fully unsupervised learning. Finally, we define a threshold on frame-wise reconstruction error for detecting anomalies.

### 3.1 OVERALL MLE-UVAD STRUCTURE

Our MLE-UVAD framework is designed to separate anomalies in the unlabelled video with a convolutional autoencoder (CAE), as shown in Fig. 2. This autoencoder serves as the reconstruction foundation (Li et al., 2023): it encodes frames into latent representations and decodes them back to the image space. First, the reconstruction loss enables accurate video reconstruction, where normal and abnormal videos typically occupy different distribution regions. Second, the MLE loss then reduces entropy in the latent space, collapsing abnormal embeddings into the dominant normal cluster. Together, these losses guarantee that normal frames are reconstructed well while abnormal frames are reconstructed poorly, creating a clear reconstruction gap. Thus, the dual-loss objective balances two effects: MSE preserves normal reconstructions, while MLE enforces distributional regularization that magnifies the gap between normal and abnormal frames.

After training the autoencoders, we measure the reconstruction quality via the Pearson Correlation Coefficient (PCC) for detection. PCC emphasizes structural variation and is invariant to global

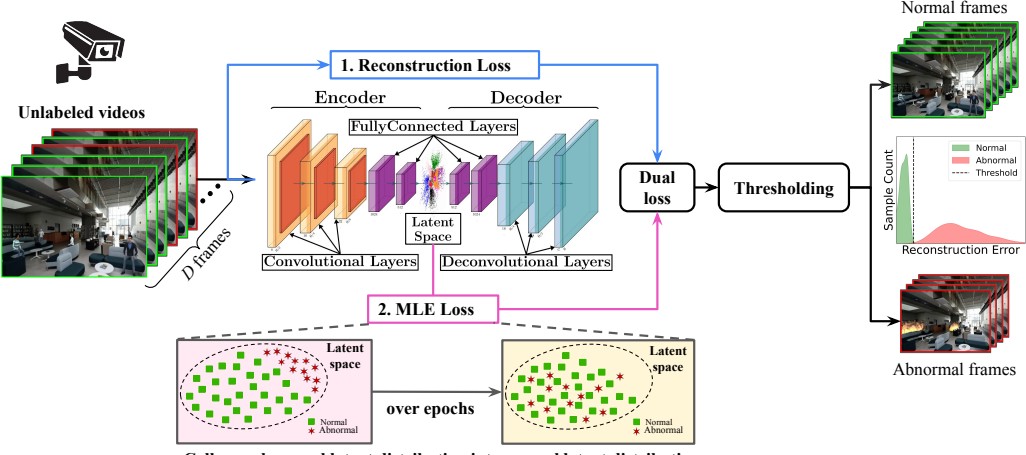

Figure 2: MLE-UVAD pipeline: unsupervised training and detection through reconstruction gap. Unlabeled videos are directly put into an autoencoder trained with a dual loss: (1) The MSE loss reconstructs all frames accurately. (2) The MLE collapses sparse abnormal embeddings toward the dense normal embedding, making anomaly reconstructions worse relative to normals. An anomaly is detected by setting a threshold on the reconstruction error.

brightness changes, making it more suitable than MSE (sensitive to scale) and cosine similarity (sensitive to shift). The PCC is calculated between original frames $\mathbf{x_i}$ and its reconstruction $\mathbf{x_i'}$:

$$\text{pcc}_i = \frac{(\mathbf{x}_i' - \bar{x}_i')^\mathsf{T}(\mathbf{x}_i - \bar{x}_i)}{||\mathbf{x}_i' - \bar{x}_i'||_2||\mathbf{x}_i - \bar{x}_i||_2}, \tag{2}$$

where $\mathbf{x}_i$ and $\bar{x}_i$ are the $i$-th original frame in the video and its mean, respectively, while $\mathbf{x}_i'$ and $\bar{x}_i'$ are the reconstructed frame and its mean. A low PCC (i.e., high reconstruction error) typically indicates the presence of an anomaly.

## 3.2 PART 1: RECONSTRUCTION LOSS $\mathcal{L}_{\text{MSE}}$

The first loss, reconstruction Loss $\mathcal{L}_{\text{MSE}}$, forces the autoencoder to learn a compact representation from entire frames of video, and reconstruct frames from it. The autoencoder with only $\mathcal{L}_{\text{MSE}}$ can usually reconstruct both normal and abnormal frames well in the unlabeled dataset if the training is fully supervised and the model is sufficiently large.

Formally, the autoencoder consists of an encoder $\mathbf{e}_\theta(\mathbf{x}_i) = \mathbf{z}_i$ and a decoder $\mathbf{d}_{\theta'}(\mathbf{z}_i) = \mathbf{x}_i'$, where $\mathbf{x}_i$ denotes an input, $\mathbf{z}_i$ denotes its latent representation, and $\mathbf{x}_i'$ denotes the its reconstruction. The model is trained to minimize the reconstruction loss over all the $N$ input samples, defined as:

$$\mathcal{L}_{\text{MSE}}(\theta, \theta') = \frac{1}{N}\sum_{i=1}^{N}\|\mathbf{x}_i - \mathbf{x}_i'\|_2 \tag{3}$$

## 3.3 PART 2: MINIMAL LATENT ENTROPY LOSS $\mathcal{L}_{\text{MLE}}$

In addition to $\mathcal{L}_{\text{MSE}}$, we introduce the *Minimal Latent Entropy* loss $\mathcal{L}_{\text{MLE}}$, which explicitly minimizes the uncertainty (spread) of the autoencoder's latent embeddings. Entropy is a fundamental concept in information theory that quantifies the uncertainty of a random variable (Dong et al., 2024). In here, we quantify this uncertainty of the latent variable $z$ via the Rényi entropy (Rényi, 1961):

$$H_\alpha(Z) = \frac{1}{1-\alpha}\ln\int_{\mathbb{R}^d}P(z)^\alpha\,dz, \qquad \alpha > 0,\ \alpha \neq 1, \tag{4}$$

where the $P(z)$ represents the probability density function of the latent variable in $d$ dimensions, and $\alpha$ is the Rényi entropy of order. In practice, we usually set $\alpha = 2$, which applies a simple,

differentiable estimator using pairwise Gaussian kernels (Chen et al., 2019; Pallewela et al., 2025). Therefore, the second order of Rényi entropy is calculated as, $H_2(X) = -\ln \int P_X(x)^2 dx$.

However, since the true probability density $P(z)$ of latent embeddings is unknown, we approximate it non-parametrically through kernel density estimation (KDE) (Liu et al., 2021). KDE places a small "bump" (kernel) at each latent variable $z_i$ and averages them, avoiding any assumption that the data follow a specific distribution. Among possible kernels, we choose the Gaussian kernel because it is smoothly differentiable, has closed-form products/convolutions (useful for our Rényi-2 entropy), and yields accurate, stable density estimates in practice. Given latent samples $\{z_i\}_{i=1}^N \subseteq \mathbb{R}^d$, the density function approximated by KDE is

$$\hat{P}(z) = \frac{1}{N} \sum_{i=1}^N \mathcal{K}_\sigma(z - z_i),$$
(5)

where $\mathcal{K}_\sigma$ is one-dimensional Gaussian kernel with bandwidth (kernel size) $\sigma$, $K_\sigma(z - z_i) = \frac{1}{\sqrt{2\pi}\,\sigma} \exp\left(-\frac{(z-z_i)^2}{2\sigma^2}\right)$.

Next, the approximated density function $\hat{P}(z)$ is placed inside the Rényi entropy function to obtain the final MLE loss. Specifically, using the Gaussian convolution identity $K_\sigma * K_\sigma = K_{\sqrt{2}\sigma}$ for 2 order, the integral reduces to our final MLE loss (full derivation in the Appendix) :

$$\mathcal{L}_{\text{MLE}}(\sigma) = -\log\left[\frac{1}{N^2} \sum_{i=1}^N \sum_{j=1}^N \mathcal{K}_{\sqrt{2}\sigma}(z_i - z_j)\right].$$
(6)

This derivation shows that the entropy can be expressed directly through pairwise similarities between latent embeddings $z$, with $\sigma$ controlling the sensitivity of the kernel. The resulting MLE loss is thus a differentiable objective that penalizes dispersion in the latent space, encouraging embeddings to concentrate and suppressing the influence of outliers.

Finally, to integrate the MLE loss into the learning process, we define the overall training objective as the weighted sum of the $\mathcal{L}_{\text{MSE}}$ and the entropy regularization term $\mathcal{L}_{\text{MLE}}$:

$$\mathcal{L} = \mathcal{L}_{\text{MSE}} + \lambda\, \mathcal{L}_{\text{MLE}}(\sigma),$$
(7)

where the $\lambda$ sets the entropy term's influence and the $\sigma$ sets how sharply pairwise latent similarities decay. By jointly minimizing these two losses, the model learns to accurately reconstruct the dominant normal patterns (driven by $\mathcal{L}_{\text{MSE}}$), while simultaneously collapsing abnormal embeddings into the normal cluster (driven by the $\mathcal{L}_{\text{MLE}}$). Therefore, normal videos are reconstructed well, whereas abnormal videos are poorly reconstructed for detection.

## 3.4 MLE LOSS INTERPRETABILITY THROUGH t-SNE VISUALIZATION

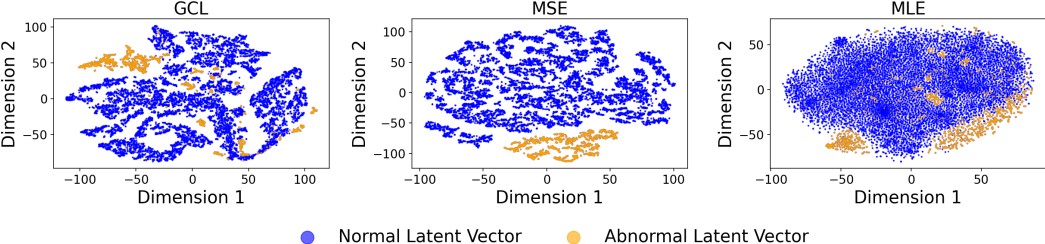

Figure 3: t-SNE visualization of latent embeddings across different methods (blue = normal, orange = abnormal). Baselines, only MSE and GCL (Zaheer et al., 2022), keep the normal and abnormal embeddings separate for better reconstruction. In contrast, our proposed MLE loss regularizes the latent distribution by collapsing the abnormal distribution into the dominant normal cluster.

To demonstrate how the MLE loss $\mathcal{L}_{\text{MLE}}$ improves normal reconstruction and degrades anomalies reconstruction, we apply t-Distributed Stochastic Neighbor Embedding (t-SNE) (Yang et al., 2025) to map the latent space (e.g., 32 dimensions) into a 2D space while preserving local relationships.

Embeddings of abnormal frames tend to constitute separate clusters in latent space for baselines (e.g.,GCL (Zaheer et al., 2022) or autoencoder with only reconstruction loss) This is expected: abnormal and normal videos differ inherently, so accurate reconstruction (mapping from latent to image) naturally places them in different latent regions. Adding the MLE loss makes the latent distribution more compact (Fig. 3: the variance along t-SNE axes decreases relative to MSE only). As a result, entropy minimization pulls the previously separated abnormal embeddings toward the dominant normal cluster, reducing uncertainty. Anomalies are reconstructed *as if they were normal*, yielding poor reconstructions. Although the MLE loss also slightly shifts normal clusters, its impact on normal reconstructions is minimal compared with its effect on anomalies. Consequently, anomalies incur higher reconstruction errors, enabling a clear reconstruction error threshold for detection.

## 3.5 ANOMALY DETECTION INDICATOR

After the dual-loss training process, we set a simple yet effective threshold on the reconstruction error. The reconstruction quality is measured by the PCC between each frame and its reconstruction (high PCC means low reconstruction error). Since our dual loss makes the autoencoder reconstruct the anomaly poorly, anomalies lie in the *lower tail* of the PCC distribution. Given the $\{\mathrm{pcc}_i\}_{i=1}^T$ across all frames in time horizon $T$, the global mean $\mu$ and standard deviation $\sigma$ as

$$\mu = \frac{1}{T} \sum_{i=1}^{T} \mathrm{pcc}_i, \qquad \sigma = \sqrt{\frac{1}{T} \sum_{i=1}^{T} (\mathrm{pcc}_i - \mu)^2}, \tag{8}$$

We apply a lower-tail threshold to detect anomalies:

$$\tau = \mu - \kappa\sigma, \qquad \text{declare } x_i \text{ anomaly if } \mathrm{pcc}_i < \tau. \tag{9}$$

We set $\kappa = 0.5$ for all scenarios because it consistently balances sensitivity to anomaly with a low false-alarm rate. When PCCs are tightly concentrated (small $\sigma$), the threshold becomes stricter; when scores are more spread out (large $\sigma$), the threshold relaxes. This variance-aware rule avoids manual threshold tuning while still reliably separating normal and abnormal frames.

## 4 EXPERIMENTS

### 4.1 EXPERIMENTAL DESIGN AND IMPLEMENTATION

**Datasets.** We evaluated our approach on three datasets, covering both synthetic and real-world scenarios under different anomaly ratios. They provide diverse challenges: Donkeycar captures real driving with sensor noise, Corridor represents real surveillance footage with the protest abnormal events, and UBnormal introduces controlled synthetic anomalies. Further details are provided below.

*Donkeycar (real-world, self-collected).* This is a real-world dataset collected with a camera on the Donkeycar self-driving platform (Viitala et al., 2021). Normal frames correspond to clean driving views, while anomalies are defined as raindrops obscuring the monitor (See Fig 10, top left). It contains 18,000 frames (6,400 pixels per frame) with an anomaly ratio of 12.5%. The moving background makes it difficult for the autoencoder to detect anomalies.

*Corridor (real-world, public).* A fixed-view surveillance dataset capturing activities in a corridor environment (Rodrigues et al., 2020). We evaluate the protest anomaly type, yielding 1,200 frames (187,200 pixels each) and an anomaly ratio of 20% (Figure 10, bottom). The main challenges are the limited training size, which makes reconstruction less stable, and the fact that anomalies manifest as crowd events, representing realistic and complex real-world anomaly scenarios.

*UBnormal (synthetic, public).* This 3D-generated anomaly detection benchmark contains multiple types of anomalies in a virtual environment (Acsintoae et al., 2022). We test the model in scene index 4 with the fire alarm anomaly type, resulting in 902 frames (187,200 pixels each) and an anomaly ratio of 50% (Figure 10, top right). The small training size and 1:1 normal–abnormal ratio make it especially challenging

**Baseline Methods.** We compare our proposed method against three representative *reconstruction-based* baseline methods: Vanilla CAE, a standard CAE trained with MSE loss that serves as an ablation, but may also reconstruct abnormal patterns, limiting detection performance. GCL combines

a convolutional autoencoder and a discriminator in a feedback loop (Zaheer et al., 2022), where the discriminator flags poor reconstructions to guide the autoencoder; and TMAE detects anomalies based on masked autoencoder (Hu et al., 2022). FUN-AD performs fine-grained anomaly scoring and enforces local consistency in patch embeddings via mutual smoothness and a memory bank (Im et al., 2025). FRD-UVAD discriminates anomalies with selective cross-attention reconstruction and a disruption mechanism that suppresses abnormal features (Tao et al., 2024).

**Implementation Details.** All models are trained with a CAE of embedding dimension 32 for 70 epochs using Adam ($5 \times 10^{-4}$ learning rate). Batch sizes: 256 for Donkeycar (18k frames), 128 for Corridor (1.2k frames), and 64 for UBnormal (902 frames). We adopt a fully unsupervised setting in which both normal and abnormal frames are present during training; training and testing share the same unlabeled data, with labels accessed only for reporting metrics. After each epoch, we record PCC and AUC to monitor performance. To study the MLE loss, we test kernel size $\sigma$ and weight $\lambda$ over $\{0.01, 0.05, 0.1, 0.5, 1.0\}$. Evaluation relies on two complementary metrics: frame-level PCC for anomaly sensitivity, AUC for overall detection accuracy.

## 4.2 EXPERIMENTAL RESULTS

Our experiments answer the following questions: (A) How does the MLE loss enhance unsupervised detection? (B) What is the impact of the hyperparameters $\sigma, \lambda$ on the detection performance? (C) What is the impact of varying anomaly ratios on the model's performance? (D) How well does our method generalize to different types of anomalies?

Table 1: AUC comparison of anomaly detectors across datasets.

| Category | Method | Donkeycar | Corridor | UBnormal |
|---|---|---|---|---|
| OCC (Semi-supervised) | DAST (Kommanduri & Ghorai, 2024) | 0.999 | 0.996 | 0.991 |
| | ROADMAP (Wang et al., 2021) | 1.000 | 0.996 | 0.991 |
| | GAN (Ravanbakhsh et al., 2017) | 0.559 | 1.000 | 0.979 |
| Fully Unsupervised | FUN-AD (Im et al., 2025) | 0.633 | 0.478 | 0.499 |
| | FRD-UVAD (Tao et al., 2024) | 0.751 | 1.000 | 0.935 |
| | TMAE (Hu et al., 2022) | 1.000 | 0.994 | 0.901 |
| | GCL (Zaheer et al., 2022) | 0.954 | 0.000 | 0.028 |
| | Vanilla Autoencoder | 0.691 | 0.787 | 0.066 |
| | MLE-Guided Autoencoder (ours) | 1.000 | 1.000 | 1.000 |

**The Effect of MLE loss on fully UVAD.** With the MLE loss ($\lambda = 1, \sigma = 0.1$), our entropy-guided autoencoder consistently outperforms fully unsupervised baselines, TMAE (Hu et al., 2022), GCL (Zaheer et al., 2022), FUN-AD (Im et al., 2025), FRD-UVAD (Tao et al., 2024), and a vanilla autoencoder with only MSE in Table 1. Moreover, it attains near-perfect AUCs ($\approx 1.0$) trained without labels, matching the best OCC results. In contrast, the vanilla autoencoder without MLE largely fails (e.g., AUC $\approx 0$ in UBnormal dataset). Our proposed MLE-guided autoencoder demonstrates robust anomaly sensitivity under various conditions.

To further explain the effect of MLE loss beyond Table 1, we plot the per-frame reconstruction quality measured with PCC across all the datasets, as shown in Fig 4. Our entropy-guided CAE maintains high PCC on normal frames (e.g., PCC $\approx 0.97$ on Corridor) and shows clear drops on abnormal frames (e.g., PCC $\approx 0.95$ on Corridor), yielding a stable reconstruction gap. This gap is large compared to per-frame variance, so the $\mu - 0.5\sigma$ threshold can be reliable for detecting anomalies. Furthermore, this gap happens because the MLE loss reduces latent entropy and concentrates embeddings on the dominant normal distribution. Abnormal embeddings are pulled toward the normal distribution, which prevents the decoder from overfitting to abnormal patterns. Abnormal inputs are reconstructed as a normal sample, producing a larger reconstruction error and lower PCC.

In contrast, the baselines fail to separate anomalies because they do not induce a reconstruction drop. Vanilla autoencoder, which minimizes global reconstruction error, often over-reconstructs both normal and abnormal frames, leaving only small reconstruction differences between them. Take the UBnormal dataset as an example in Fig 4, PCC of normal and abnormal frames remains

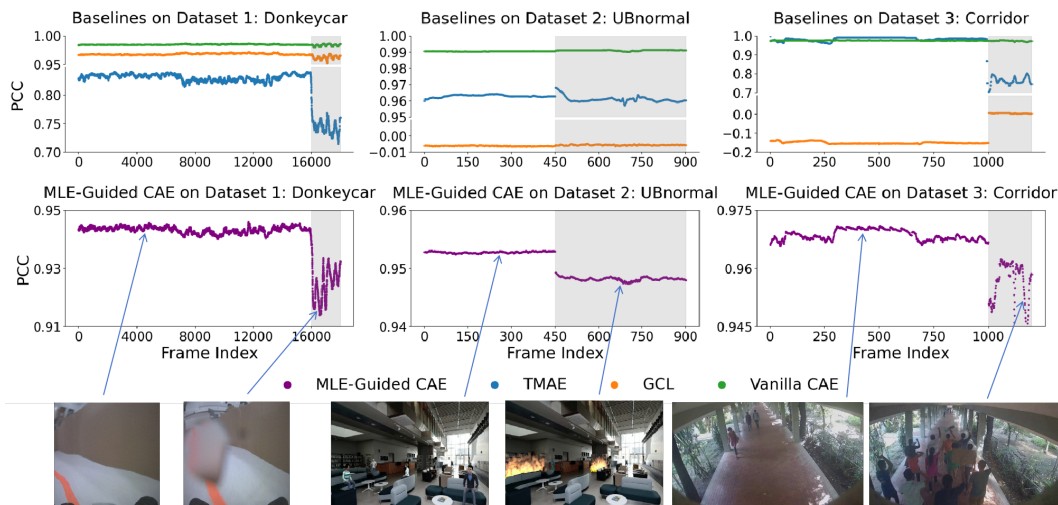

Figure 4: Unsupervised anomaly detection performance across three benchmarks. Top row: baselines (TMAE, GCL, Vanilla CAE). Bottom row: our MLE-Guided CAE. Columns: DonkeyCar, UBnormal, Corridor. Our method shows a clear normal–anomaly separation of the PCC value.

very high for the baselines ($\approx 0.96$ for TMAE; $\approx 0.99$ for the vanilla CAE), effectively hiding anomalies. Meanwhile, GCL exhibits unstable behavior across datasets, sometimes even collapsing to near-random performance (e.g., PCC $\approx 0$ in the UBnormal and Corridor dataset).

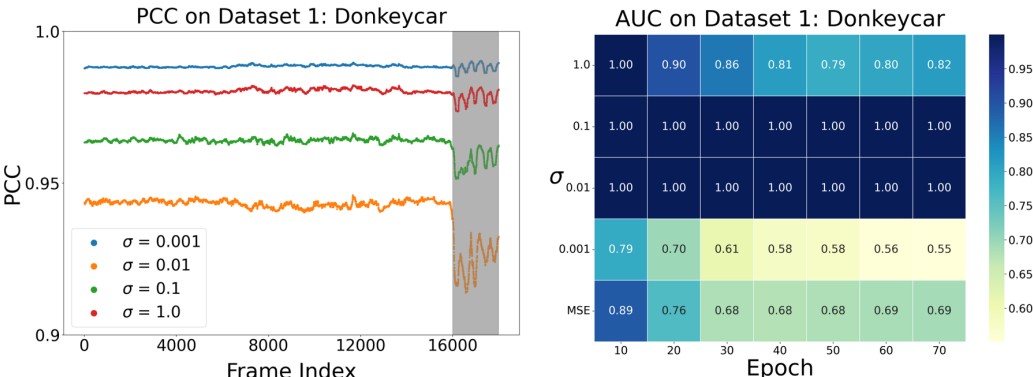

Figure 5: Effect of kernel size $\sigma$ in the MLE loss. Left: PCC trajectories for different $\sigma$ values; gray = anomaly. Right: AUC heatmap across epochs. Mid-range $\sigma$ (0.01–0.1) yields stable reconstructions and high AUC, while very small or large $\sigma$ degrades detection.

**Effect of Kernel Size $\sigma$ on Detection Performance.** We evaluate detection performance across ranges of the two hyperparameters: kernel size $\sigma$ and its weight ratio $\lambda$ of MLE. The kernel size $\sigma$ controls the smoothness of the Gaussian kernel: a very small $\sigma$ makes kernels overly sharp, whereas a very large $\sigma$ makes them nearly flat. At either extreme $\sigma$, such as 1 or 0.001, no clear reconstruction gap exists (See blue and red PCC in Fig. 5). Therefore, it is challenging to establish a threshold to distinguish between anomalies. But for intermediate $\sigma$ values, a clear and easily detectable reconstruction gap emerges (See green and orange PCC in Fig. 5). Moreover, extreme $\sigma$ values lead to unstable training and behave similarly to a vanilla autoencoder, whereas a moderate $\sigma$ yields stable training and robust separation, as shown in the AUC score heatmap in Fig. 5.

These behaviors happen because extremely small or large $\sigma$ makes the MLE loss non-informative, causing training to transfer to the reconstruction loss, and preventing abnormal embeddings from collapsing toward the normal manifold. From a kernel density estimation (KDE) perspective, our

loss induces *pairwise Gaussian affinities* for embeddings,

$$w_{ij} = \exp\left(-\frac{(z_i - z_j)^2}{4\sigma^2}\right), \tag{10}$$

and the gradient on a latent embedding variable $z_k$ is

$$\nabla_{z_k}\mathcal{L} \propto -\frac{1}{\sigma^2}\sum_j (z_k - z_j)\, w_{kj}, \tag{11}$$

Therefore, each latent variable is pulled toward a kernel-weighted average of its neighbors. When $\sigma$ is *too small*, the kernels are extremely sharp and $w_{ij} \approx 0$. As a result, the gradient of the loss in equatio (11) becomes negligible for most $z_k$, providing no clear direction to concentrate normals (training stalls). When $\sigma$ is *too large*, the Gaussian kernels are nearly flat. Since $w_{ij} \approx 1$, the loss cannot tell "near" from "far." Each latent embedding $z_k$ is uniformly attracted toward all the neighborhood, not toward high-density regions. That destroys the mechanism to cluster normals while repelling anomalies. Therefore, only a moderate $\sigma$ preserves meaningful weights between near and far embedding pairs to produce informative gradients.

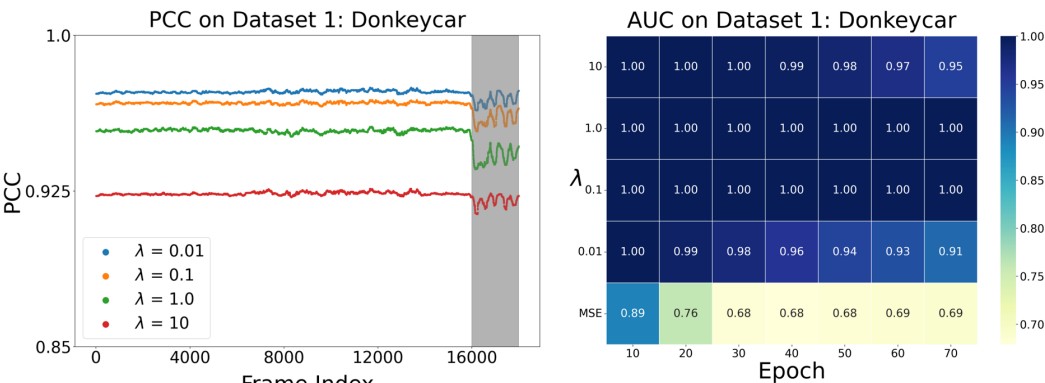

Figure 6: Effect of MLE loss weight $\lambda$ on anomaly detection. Left: PCC trajectories for different $\lambda$ values; gray = anomaly. Right: AUC heatmap across epochs. Moderate $\lambda$ (0.1–1.0) achieves the best trade-off between stable reconstruction and clear anomaly sensitivity.

**Effect of Loss Weight $\lambda$ on Detection Performance**. The weight $\lambda$ scales the MLE loss in the total loss $\mathcal{L} = \mathcal{L}_{\text{MSE}} + \lambda\,\mathcal{L}_{\text{MLE}}$. When $\lambda$ is too small (e.g., 0.01), the entropy regularization is largely ignored, making training nearly MSE-only and weakening anomaly sensitivity; this leads to reduced PCC drops on abnormal frames and lower AUC, shown in Fig. 6. Conversely, when $\lambda$ is too large (e.g., 10), the entropy term dominates, over-regularizing the embeddings, which degrades reconstruction quality for both normal and abnormal frames, causing a drop in AUC. In contrast, mid-range values of $\lambda$ (e.g., 0.1–1.0) provide the best balance: reconstructions of normal frames remain stable, while abnormal embeddings are sufficiently collapsed to yield clear PCC separation and consistently high AUC across epochs. In short, $\lambda$ values from 0.001 to 1 and $\sigma$ values from 0.01 to 0.1 consistently achieve near-optimal AUC across all epochs. In practice, we suggest using any combination within these ranges for reliable anomaly detection.

**Effect of the Anomaly Ratio on Detection Performance**. Furthermore, we evaluated our model's performance across anomaly ratios ranging from 10% to 60% on all three datasets. When the anomaly ratio is below 40%, the model maintains strong performance with AUC scores around 0.9. However, once the ratio exceeds 50%, the AUC drops sharply to approximately 0.7 (see Fig. 7). This decline occurs because a larger anomaly ratio shifts the dominant latent-space cluster from normal to abnormal. Since MLE reduces overall entropy by pulling embeddings toward the main concentration, the normal cluster is dragged toward the abnormal one. As a result, normal frames are reconstructed poorly, and the reconstruction gap between normal and abnormal frames becomes difficult to observe. Nonetheless, in real-world camera systems, anomaly ratios above 50% are rare, and we expect our model to be consistently effective with anomaly ratios below 40%.

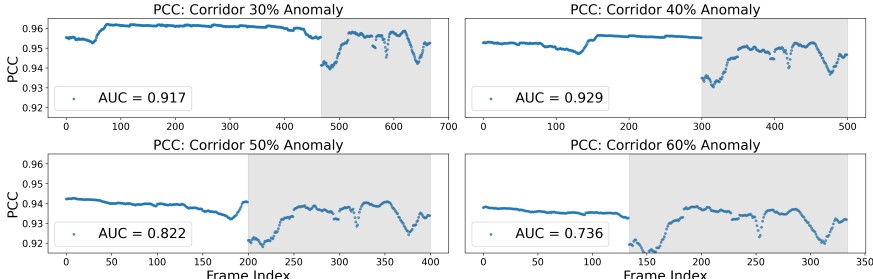

Figure 7: Impact of anomaly ratios on detection performance for the Corridor dataset. The subplots show the PCC trajectories at different anomaly ratios, with the gray regions indicating anomalies.

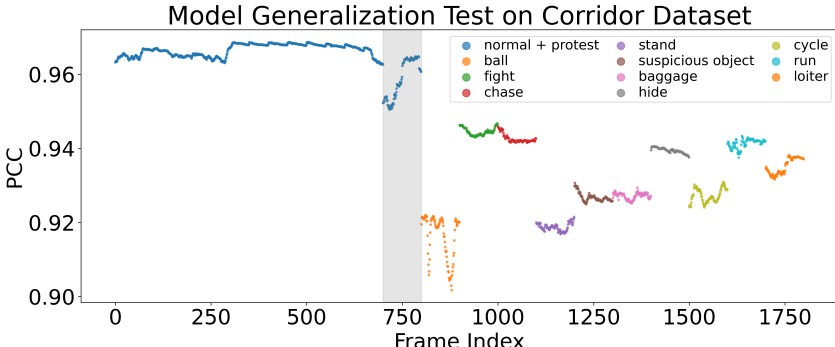

Figure 8: Test of generalization across 10 different anomalies on the Corridor dataset. The model is trained on normal with only protest data (gray area: protest) and is tested on the other 10 anomalies.

**Evaluating Generalization Capability Across Different Anomalies.** We evaluate the generalization ability by training our model on an unlabeled video containing normal scenes and a single anomaly type — and testing it on the other, unseen anomalies. These unseen anomalies show a clear PCC drop compared to normal frames, indicating that they can be effectively detected (see Fig. 8). That is because the MLE loss encourages the model to learn the compact normal manifold by emphasizing the normal data distribution and preventing overfitting to the known anomaly during training. During inference, other unseen anomalies still deviate from this learned manifold and therefore cannot be reconstructed well, leading to a significantly lower PCC. Since normal frames maintain high PCC, the resulting gap enables reliable anomaly detection. Therefore, the MLE loss also supports strong generalization to anomaly types not observed during training.

## 5 CONCLUSION

This paper introduced a single-scene, fully unsupervised VAD setting and an entropy-guided autoencoder (MLE-UVAD) that adds minimal latent entropy to the reconstruction loss. This dual loss collapses sparse anomalous embeddings toward the dominant normal cluster, yielding a stable reconstruction-error gap that enables reliable, high-AUC detection in three datasets. We further analyze the effects of key hyperparameters, different anomaly ratios, and the model's ability to generalize to unseen anomaly types. Our future work will deploy MLE-UVAD as a privacy-preserving data-cleaning method that outputs high-confidence normal/abnormal masks for supervised training without sharing the raw video. Also, we will extend this approach to multi-camera/multi-scene settings through domain adaptation and camera-aware priors while incorporating temporal modeling to curb false alarms from transient events.

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

## APPENDIX

**Appendix Contents**

## A  MINIMAL LATENT ENTROPY LOSS DERIVATION

Entropy is a fundamental concept in information theory that quantifies the uncertainty of a random variable. For discrete variables, Shannon entropy provides a natural measure of information content. However, latent embeddings in our setting are treated as samples from an underlying continuous distribution, rather than categorical outcomes. Applying discrete Shannon entropy directly would require binning or counting, which is highly sensitive to batch size and suffers from large bias when the number of samples is small. In contrast, differential entropy provides a principled way to measure uncertainty in continuous spaces. We adopt the Rényi entropy framework, which generalizes Shannon entropy and allows flexible order parameters $\alpha$. In particular, we use the second-order Rényi entropy ($\alpha = 2$), as it naturally integrates with kernel density estimation (KDE), admits a closed-form pairwise kernel formulation, and has been widely applied in information-theoretic learning. This choice provides both numerical stability and differentiability, making it well-suited for deep learning optimization.

We begin with the definition of Rényi entropy of order $\alpha$:

$$H_\alpha(X) = \frac{1}{1-\alpha} \log \int P_X(x)^\alpha dx. \qquad (12)$$

For $\alpha = 2$, this simplifies to the second-order Rényi entropy:

$$H_2(X) = -\log \int P_X(x)^2 dx. \tag{13}$$

Since the true distribution $P_X(x)$ is unknown, we approximate it using kernel density estimation (KDE). Among possible kernels, the Gaussian kernel is the most common choice due to its smoothness, analytic properties, and ability to approximate arbitrary densities. Gaussian kernel is expressed as $\mathcal{K}_\sigma(x) = \frac{1}{\sqrt{2\pi}\sigma} \exp\left(-\frac{x^2}{2\sigma^2}\right)$, with bandwidth $\sigma$ controls the sensitivity of the kernel. Formally, the empirical density is approximated as

$$\hat{P}(x) = \frac{1}{N} \sum_{i=1}^{N} \mathcal{K}_\sigma(x - x_i). \tag{14}$$

Substituting $\hat{P}(x)$ into the entropy expression gives

$$H_2(X) = -\log \int \left[\frac{1}{N} \sum_{i=1}^{N} \mathcal{K}_\sigma(x - x_i)\right]^2 dx \tag{15}$$

$$= -\log \int \frac{1}{N^2} \sum_{i=1}^{N} \sum_{j=1}^{N} \mathcal{K}_\sigma(x - x_i) \mathcal{K}_\sigma(x - x_j) \, dx. \tag{16}$$

The key simplification comes from the convolution property of Gaussian kernels:

$$\int \mathcal{K}_\sigma(x - x_i) \mathcal{K}_\sigma(x - x_j) \, dx = \mathcal{K}_{\sqrt{2}\sigma}(x_i - x_j).$$

Applying this result, the entropy becomes

$$H_2(X) = -\log \left[\frac{1}{N^2} \sum_{i=1}^{N} \sum_{j=1}^{N} \mathcal{K}_{\sqrt{2}\sigma}(x_i - x_j)\right]. \tag{17}$$

Now, considering the embeddings $\mathcal{E} = \{e_i\}_{i=1}^{N}$ from the autoencoder latent space, the latent entropy is

$$H_2(\mathcal{E}) = -\log \left[\frac{1}{N^2} \sum_{i=1}^{N} \sum_{j=1}^{N} \frac{1}{\sqrt{4\pi}\sigma} \exp\left(-\frac{(e_i - e_j)^2}{4\sigma^2}\right)\right]. \tag{18}$$

This expression shows that entropy can be computed entirely from pairwise similarities between embeddings. Large embedding distances (often between abnormal and normal samples) yield small kernel values and contribute little to the sum, while small distances (within the normal cluster) dominate.

As the MLE loss decreases during backpropagation, the sum of kernel contributions increases, indicating that more pairwise embedding distances are concentrated near zero. In this process, the originally large distances between abnormal embeddings and normal embeddings are affected the most, being pulled closer to zero. This effectively collapses the abnormal embedding distribution into the normal embedding cluster, leading the decoder to focus primarily on reconstructing the normal data manifold. From the essence perspective, entropy fundamentally measures the uncertainty of a distribution. Minimizing the MLE loss reduces the uncertainty of the embedding distribution, making it more concentrated. Consequently, abnormal embeddings are absorbed into the dominant normal pattern, further collapsing the abnormal distribution and ensuring that the decoder emphasizes the reconstruction of normal patterns.

To integrate the MLE loss into the learning process, we define the overall training objective as the weighted sum of the standard reconstruction loss and the entropy regularization term:

$$\mathcal{L} = \mathcal{L}_{\text{MSE}} + \lambda \, \mathcal{L}_{\text{MLE}}(\sigma), \tag{19}$$

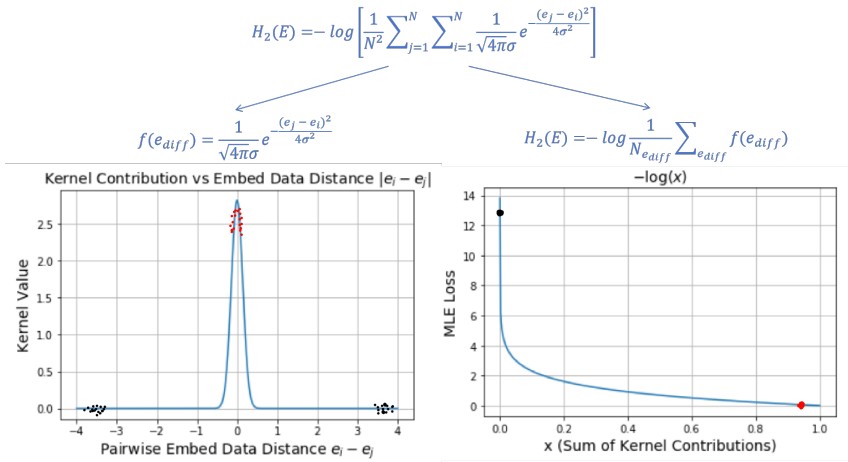

Figure 9: Two-part interpretation and visualization of the Minimal Latent Entropy (MLE) loss. This diagram decomposes the second-order Rényi entropy loss into two conceptual components. The top expression defines the MLE loss as the negative logarithm of the information potential, computed via all pairwise embedded data distances using a Gaussian kernel. The left subplot illustrates the kernel value $\mathcal{K}(e_i - e_j)$ as a function of the embedded data distance. Red points indicate small data distances (high similarity), contributing strongly to the kernel sum; black points represent large data distances (low similarity), typically stand for the distances between outliers and normal data, and contribute negligibly. The right subplot visualizes how the aggregated kernel similarity is compressed through the $-\log(x)$ function to produce the final MLE loss. As the distances grow (caused by outliers), the kernel sum drops toward zero, leading to a sharp increase in entropy. This visual demonstrates that MLE encourages tightly clustered error distributions and naturally suppresses the influence of outliers by down-weighting their pairwise contributions in the information potential.

where $\mathcal{L}_{\text{MSE}}$ is the mean squared reconstruction error, $\mathcal{L}_{\text{MLE}}(\sigma)$ is the minimum latent entropy loss parameterized by kernel size $\sigma$, and $\lambda$ is a balancing coefficient. Here, $\sigma$ controls the sensitivity of the Gaussian kernel in measuring pairwise embedding similarities, while $\lambda$ governs the relative influence of the entropy term compared to the reconstruction term. By jointly minimizing these two losses, the model learns to accurately reconstruct the dominant normal patterns (driven by the MSE term), while simultaneously collapsing abnormal embeddings into the normal cluster (driven by the MLE term). This synergy ensures that normal videos are reconstructed well, whereas abnormal videos are poorly reconstructed, thereby magnifying the residual gap and yielding a reliable signal for anomaly detection.

## B  TABLE SUMMARY OF STRENGTHS AND WEAKNESSES OF VAD

| Mode | Labeling Cost | Limitations |
|---|---|---|
| Fully Supervised | Very High (frame-level labels for all videos) | Impractical for large-scale |
| Weakly Supervised | High (video-level labels, full video review) | Hard to localize anomalous snippets |
| Unsupervised / OCC | Medium (normal-only videos) | Fails with unseen normal behaviors; sensitive to distribution shifts |
| Fully Unsupervised | Zero (no labels at any level; raw video as input) | May require adaptation for multi-camera settings |

Table 2: Comparison of anomaly detection approaches in terms of labeling cost and main limitations. Fully supervised and weakly supervised methods rely on extensive manual annotation, while previous unsupervised/OCC approaches still assume clean normal-only data and struggle with distribution shifts. In contrast, our proposed completely unsupervised setting requires no labels and operates directly on raw video, making it more scalable to real-world deployment.

## C  BASELINE EXPLANATION

**Vanilla CAE.**  A standard convolutional autoencoder trained solely with mean squared error (MSE) reconstruction loss. This method aims to minimize pixel-wise errors and serves as a strong deep learning baseline method. However, without additional constraints, it may inadvertently reconstruct both normal and abnormal patterns, limiting anomaly detection performance.

**GCL.**  Generative Cooperative Learning (GCL) integrates a convolutional autoencoder (CAE) with a multilayer perceptron (MLP) discriminator in a cooperative scheme. The autoencoder aims to reconstruct normal data, while the discriminator identifies poorly reconstructed samples and feeds this information back to the generator. Through this loop, the model gradually suppresses abnormal reconstructions and enhances its focus on the dominant normal distribution, making GCL an effective baseline for anomaly detection.

**TMAE.**  Temporal Masked Auto-Encoding (TMAE) is a recent unsupervised anomaly detection framework that adapts the masked autoencoder paradigm to video data. Instead of processing entire frames, it represents events as spatial-temporal cubes (STCs) built from consecutive foreground patches. Half of the patches in each STC are masked along the temporal dimension, and a Vision Transformer (ViT) is trained to reconstruct the missing patches from the visible ones. Because normal patterns are frequent and predictable, their masked patches can be reconstructed accurately, while rare and irregular anomalies yield larger prediction errors. These prediction errors, optionally combined with optical-flow-based motion clues, are aggregated as anomaly scores, enabling effective detection in a fully unsupervised manner.

**FUN-AD.**  FUN-AD is a fully unsupervised anomaly detection framework based on feature uncertainty modeling. This method extracts local patch-level embeddings from the input image and maintains structural consistency between spatial neighborhoods through a mutual information-based mutual smoothness loss. Simultaneously, it utilizes a dynamically updated feature memory bank to determine the nearest neighbor of each patch, generating soft pseudo-labels that aggregate high-confidence normal patterns and suppress noise and unstable features. With its "local structural constraints + feature density guidance" design, FUN-AD can distinguish between normal and anomalous patterns without feature priors or supervised labels.

**FRD-UVAD.**  Feature Reconstruction with Disruption for Unsupervsied video anomaly detection (FRD-UVAD) is a feature-level method for fully unsupervised video anomaly detection. It improves the quality of pseudo-labels and enhances the discriminative power of representations through a joint optimization mechanism of "reconstruction + perturbation". The model cascades multi-head self-attention and cross-segment attention (MSA+MCA) to reconstruct the spatiotemporal features of adjacent video segments, enabling consistent and predictable feature reconstruction of normal events both locally and across time dimensions. Simultaneously, a Latent Anomaly Memory (LAB) is introduced to continuously collect high-confidence anomaly features, and a dynamic margin "perturbation loss" is used to proactively weaken the model's reconstruction ability for these samples, thereby amplifying the reconstruction differences between normal and anomalies. Based on the reconstruction error, the model generates self-supervised pseudo-labels, and an independently trained auxiliary scorer further filters out early noisy pseudo-labels, significantly improving convergence quality.

## D  DATASET SAMPLES WITH DESCRIPTIONS

| Dataset | # Dimensions | # Instances | Anomaly ratio ($\rho$) |
|---------|--------------|-------------|------------------------|
| Donkeycar | 6400 | 18000 | 0.125 |
| UBnormal | 187200 | 902 | 0.5 |
| Corridor | 134400 | 1200 | 0.2 |

Table 3: Statistics of the experimental datasets. "# Dimensions" denotes the per-frame feature dimensionality (i.e., flattened pixel count), "# Instances" is the total number of frames, and "Anomaly ratio ($\rho$)" indicates the proportion of abnormal frames in each dataset.

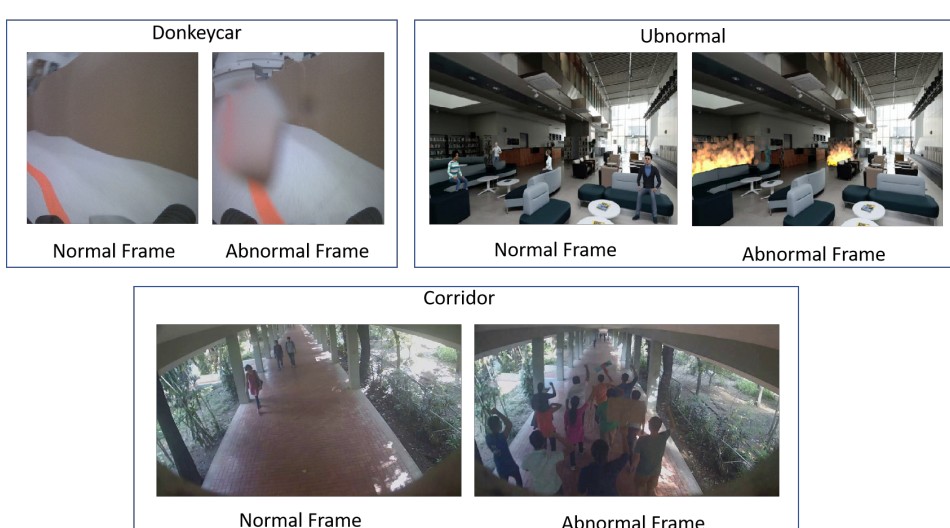

Figure 10: Examples of normal and abnormal frames from the three datasets used in our experiments: Donkeycar (top left), UBnormal (top right), and Corridor (bottom).

# E ADDITIONAL SENSITIVITY ANALYSIS OF $\sigma$ AND $\lambda$

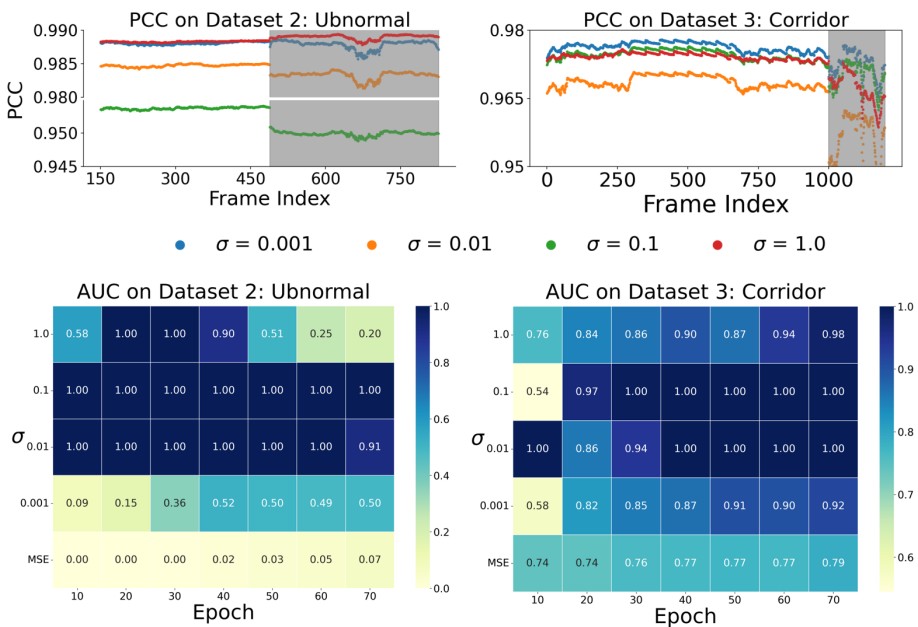

Figure 11: Effect of kernel size $\sigma$ on UBnormal and Corridor datasets. Mid-range $\sigma$ (0.01–0.1) yields clearer PCC drops on anomalies, therefore leading to high AUC scores. Extreme values (0.001 or 1.0) degrade detection performance, leading to low AUC scores.

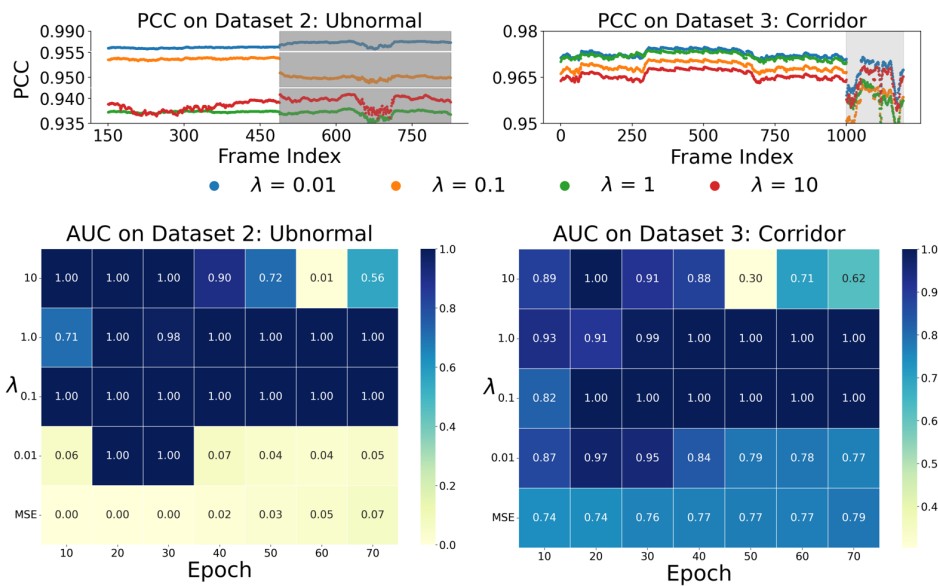

Figure 12: Effect of MLE loss weight $\lambda$ on UBnormal and Corridor datasets. Moderate $\lambda$ (0.1–1.0) balances normal reconstruction and anomaly sensitivity, achieving high AUC scores. whereas too small (0.01) or too large (10) weakens detection, leading to low AUC scores.

# F ROBUSTNESS EVALUATION ACROSS ANOMALY RATIOS, DIFFERENT SCENARIOS AND SSIM-BASED RECONSTRUCTION

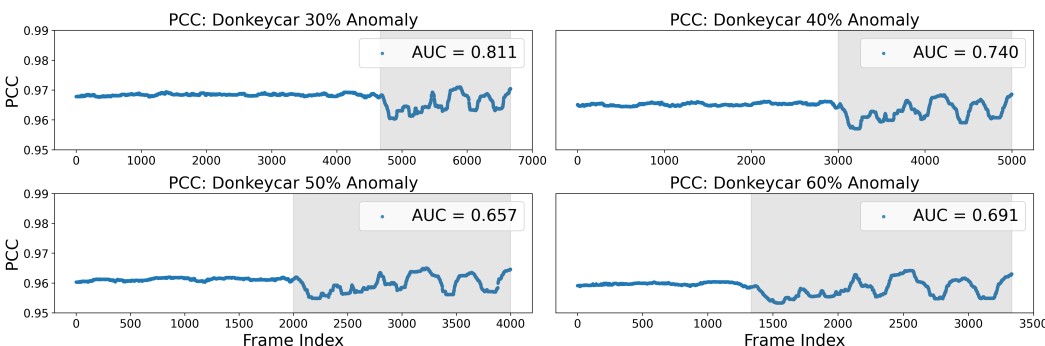

Figure 13: The effect of different anomaly ratios on detection for the Donkeycar dataset. As the anomaly ratio increases from 30% to 60%, the PCC curve of anomaly gradually flattens out, and the AUC also decreases, indicating that the higher the anomaly ratio makes detection more challenging.

Table 4: AUC performance under different anomaly ratios on three datasets.

| Dataset | 30% | 40% | 50% | 60% |
|---|---|---|---|---|
| Donkeycar | 0.811 | 0.740 | 0.657 | 0.691 |
| UBnormal | 1.000 | 1.000 | 1.000 | 0.900 |
| Corridor | 0.917 | 0.929 | 0.822 | 0.736 |

Table 5: AUC scores of different anomaly scenarios for the Corridor dataset.

| Anomaly type | MLE-Guided CAE | FRD-UVAD | FUN-AD | GCL |
|---|---|---|---|---|
| protest | 1.000 | 1.000 | 0.579 | 0.999 |
| ball | 1.000 | 0.992 | 0.474 | 0.926 |
| stand | 1.000 | 0.948 | 0.431 | 0.680 |
| baggage | 0.964 | 0.998 | 0.497 | 0.007 |
| suspicious object | 0.881 | 1.000 | 0.489 | 0.381 |

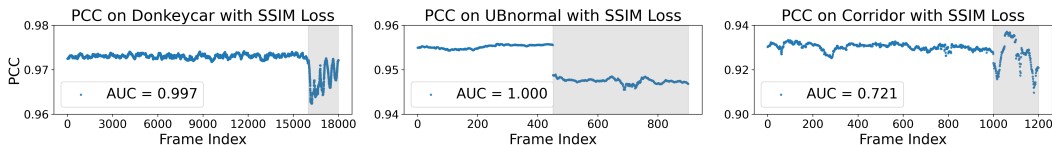

Figure 14: The effect of the SSIM + MLE loss on detection for the Donkeycar, UBnormal, and Corridor datasets. The SSIM loss achieved high AUC on Donkeycar and UBnormal dataset, but showed some performance degradation on Corridor data.

