# OpenReview forum: "MLE-UVAD: Minimal Latent Entropy Autoencoder for Fully Unsupervised Video Anomaly Detection"
_ICLR.cc/2026/Conference — Submitted to ICLR 2026_

### Official Review · Reviewer_HpRQ · 2025-10-30

**Soundness:** 2
**Presentation:** 2
**Contribution:** 2
**Rating:** 4
**Confidence:** 4

**Summary:**

This paper presents a new approach to fully unsupervised VAD, called MLE-UVAD. The method uses a convolutional autoencoder and introduces a novel MLE loss in combination with reconstruction loss to address the problem of detecting anomalies in a single-scene video setting. The proposed system aims to detect abnormal events in videos without any labeled data, making it suitable for privacy-sensitive applications. Through experiments on several datasets, the authors show that their method outperforms various baseline models, including those with one-class classification and other unsupervised approaches.

**Strengths:**

1. The MLE loss is a novel approach to guide the autoencoder to collapse anomalous embeddings into the dominant normal cluster. This entropy-guided regularization improves the reconstruction gap, which is crucial for detecting anomalies in the absence of labels.
2. The proposed method operates in a fully unsupervised setting, meaning no labeled data is required during training, which is a significant advantage for real-world scenarios where labeled data is scarce or unavailable.
3. The method is tested on three diverse datasets, including real-world data and synthetic data, showing robust performance across various anomaly types and settings, outperforming both fully supervised and unsupervised baseline methods.

**Weaknesses:**

1. The paper uses the same unlabelled data for both training and testing, which leads to potential information leakage. It also tracks AUC with true labels, which can bias results.
2. The MLE loss has high computational complexity and relies on sensitive hyperparameters like σ/λ, but there's no efficiency data such as FPS or memory usage.
3. The experiments use smaller datasets and report good AUC scores, even with a high anomaly ratio. It is recommended to test on larger and more diverse benchmark datasets, including more metrics such as precision, recall, and F1 score.

**Questions:**

1. How does the method perform when the assumption of dominant normal samples is violated in real-world scenarios? Could you provide more detailed results for cases with a higher proportion of anomalies?
2. Could you clarify how the hyperparameters, particularly σ and λ, are selected in practice? Are there any automated strategies or guidelines for setting these values, especially when no labeled data is available?

---

> ### Author Response · Authors · 2025-11-22
> **Response to Reviewer HpRQ**
>
> Thanks for all the insightful opinions! We address these concerns one by one below.
>
> ### Weakness 1: "The paper uses the same unlabelled data for both training and testing, which leads to potential information leakage. It also tracks AUC with true labels, which can bias results."
> Sorry for not clarifying this clearly.
> Although the same *unlabelled video* is used for both training and testing, **no label information is accessed during training**. All frames are treated equally, and the model has no knowledge of which frames are normal or abnormal. Ground-truth labels are **used only during evaluation** to compute the AUC; without labels, model validation would not be possible. This setup is standard in fully unsupervised VAD that relies on unlabelled raw videos for training.
>
> We appreciate the reviewer’s perspective and have strengthened our evaluation accordingly. We now train on an unlabelled video containing one anomaly (e.g., protest) and **test on a different, previously unseen anomaly** (e.g., chasing). As shown in Figure 8 in the revised manuscript, the PCC gap persists for unseen anomalies, confirming the model’s generalization ability.
>
> ### Weakness 2: "The MLE loss has high computational complexity and relies on sensitive hyperparameters like σ/λ, but there's no efficiency data such as FPS or memory usage."
> Thanks for pointing this out. We would like to clarify that the computational cost of the MLE loss is actually quite low:
> * **The MLE term operates only in the latent space**. The dimensionality of dataset is already drastically reduced compared to the original image space. As a result, the entropy-minimization operation is lightweight and efficient during training.
> * **The MLE loss is used only during training**. At inference time, the model performs anomaly detection without the MLE term, and the runtime cost is identical to any standard decoder-based methods. Therefore, the inference cost (FPS, memory) is the same as the standard reconstruction-based VAD models.
>
> We will clarify this in the revised manuscript and add a brief discussion on computational efficiency.
>
> ### Weakness 3: "It is recommended to test on larger and more diverse benchmark datasets"
> We appreciate this suggestion. Our method is specifically designed for a fully unsupervised, single-scene setting, which requires one continuous video from a single camera, mixed normal/abnormal frames, and frame-level ground truth available solely for evaluation. Many large-scale benchmarks do not satisfy these constraints. Nevertheless, if reviewers recommend a dataset that matches this setting, we are happy to include additional experiments.
>
>
> ### Question 1: " Could you provide more detailed results for cases with a higher proportion of anomalies?"
> Thanks for your helpful suggestion, which indeed makes our experiments more comprehensive. We performed additional experiments with varying anomaly ratios from 10% to 60% across all datasets, as shown in the table below. Performance is stable up to 40% anomalies with a relatively high AUC of around 0.9.
> However, when anomalies exceed 50%, the MLE term loses guidance because the “dominant cluster” is no longer normal-dominated. This performance is consistent with our theoretical assumption that our model works when anomalies are naturally sparse.
>
> |   | 20% | 30% | 40% | 50% | 60% |
> | ------------- |:-------------:| :-------------:|  :-------------:|  :-------------:| :-------------:|
> | Donkeycar    | 1.000 | 0.811 | 0.740 | 0.657 | 0.691 |
> | Corridor     | 1.000 | 0.917 | 0.929 | 0.822 | 0.736 |
> | UBnormal | 1.000 | 1.000 | 1.000 | 1.000  | 0.900|
>
> At the same time, anomaly ratios above 50% are uncommon in real-world surveillance, where anomalies are generally rare. While such extreme cases fall outside our scope, we appreciate the reviewer’s observation and have added a detailed discussion along with the requested PCC figures in the revised manuscript!
>
> ### Question 2: "Could you clarify how the hyperparameters, particularly σ and λ, are selected in practice? Are there any automated strategies for setting these values?"
>
> That’s an important question for practical deployment! Our model is robust across a wide range of σ and λ values, as shown in Figures 5 and 6. In particular, λ values from 10−3 to 1 and σ values from 0.01 to 0.1 consistently achieve near-optimal AUC across all epochs. Because our model's performance is stable throughout these ranges, any combination within them works reliably, and the method does not require fine-grained hyperparameter tuning in realistic deployment.

---

> > ### Author Response · Authors · 2025-11-26
> >
> > Dear Reviewer HpRQ,
> >
> > We would like to kindly check whether our responses have fully addressed your concerns. If you are satisfied with the revisions and feel that our changes have strengthened the submission in line with your suggestions, we would greatly appreciate an update to the review scores.
> >
> > If any additional clarification is needed, please let us know—we would be happy to provide further details.
> >
> > Wishing you a happy Thanksgiving!

---

> > > ### Comment · Reviewer_HpRQ · 2025-11-27
> > > **response to the reviewer**
> > >
> > > The response from the author is comprehensive and successfully addressed part (most) of my technical conserns about this manuscript, and I am willing to support its current version.

---

> > > > ### Author Response · Authors · 2025-11-27
> > > >
> > > > Thank you for your supportive feedback on our paper and for indicating a positive assessment. We greatly appreciate your time and consideration. When you have a moment, could you also update the score so that it reflects this assessment? It currently still shows the original evaluation of point 4.

---

### Official Review · Reviewer_BmTr · 2025-10-30

**Soundness:** 3
**Presentation:** 3
**Contribution:** 3
**Rating:** 6
**Confidence:** 4

**Summary:**

This paper proposes a novel conceptual understanding of unsupervised video anomaly detection in single-scene settings via frame reconstruction. In such task, the common practice in the field has been to train autoencoder-style models to reconstruct normal videos and use the difference between the reconstructed and the ground truth frames to identify anomalous frames. The assumption is that the model learns to reconstruct normal frames very well, while struggling to reconstruct anomalous frames due to the data imbalance in the training set (consisting of more normal videos than abnormal, or only normal videos). The paper argues that clustering together the latent representation of normal and abnormal frames alike via the proposed Minimal Latent Entropy (MLE) limits the model's capability to adapt its latent distribution to both the distributions of normal and abnormal frames, leading to much more evident differences in the reconstructed anomalous frames. The experiments presented on three standard benchmark datasets show that such approach is effective.

**Strengths:**

* The paper is written clearly and concisely. The figures are informative and the ablation studies extensive.
* The conceptual design of the MLE loss is well thought and the mathematical formulation of MLE appears to be correctly formulated and presented.

**Weaknesses:**

* The experimental settings for the main table are inconsistent with common practice in literature. In lines 313-314 the authors write that the models are trained and evaluated only on a single anomaly type ("protest") of the Corridor dataset. This is not necessarily incorrect due to the focus on single-scene anomaly detection, the paper's contribution would benefit from similar experiments conducted on the rest of the anomaly types in the same scene (i.e. "chasing", "fighting", "suspicious object", etc...), either jointly or separately. Similar considerations hold for the experiments on the UBnormal dataset, of which the authors only use the "fire alarm" anomaly type (lines 318-319).

**Questions:**

* What is the combined effect of MLE with other reconstruction loss functions such as SSIM?

---

> ### Author Response · Authors · 2025-11-22
> **Response for Reviewer BmTr**
>
> We sincerely thank the reviewer for the constructive and detailed feedback! Below, we addressed the main weakness and question, and we will incorporate the corresponding improvements in the revised version.
>
> ### Weakness 1: "The paper's contribution would benefit from similar experiments conducted on the rest of the anomaly types in the same scene (i.e. "chasing", "fighting", "suspicious object", etc...), either jointly or separately."
>
> Thanks for pointing out this detailed question! This week, we conducted different anomaly types under our fully unsupervised setting, such as "protest", "ball", "stand", and "baggage". Below, we report the AUC results for our method compared with other baselines:
> | Anomaly types (15% anomaly ratio)  | AUC of our method | AUC of baseline GCL | AUC of baseline FRD-UVAD | AUC of baseline FUN-AD
> | ------------- |:-------------:| :-------------:| :-------------:| :-------------:|
> | "protest"      | 1.000 | 0.999 | 1.000  | 0.579 |
> | "ball"     | 1.000    | 0.926 | 0.992| 0.474|
> | "stand"     | 1.000   | 0.680 | 0.948 | 0.431 |
> | "baggage"     | 0.964 |  0.007 | 0.998 | 0.497|
>
> Our approach still demonstrates high ACU consistently. The baseline, FRD-UVAD, shows similarly strong performance, confirming the difficulty of these datasets.
> In contrast, the other baselines show *large fluctuations* across anomaly types, highlighting the importance of robustness under fully unsupervised conditions. These results are all included in Table 5 in our revised manuscript.
>
> ### Question 1: "What is the combined effect of MLE with other reconstruction loss functions such as SSIM?"
>
> Thanks for this insightful question! We have evaluated the combination of MLE with SSIM on all three datasets. In general, SSIM performs worse than MSE. For example, on the DonkeyCar dataset, the AUC with SSIM + MLE dropped to 0.721. The corresponding PCC plots are provided in Figure 14 in the revised manuscript.
>
> |   | Donkeycar | UBnormal | Corridor |
> | ------------- |:-------------:| :-------------:|  :-------------:|
> | SSIM + MLE    | 0.997 | 1.000 | 0.721 |
> | MSE + MLE     | 1.000 | 1.000 | 1.000 |
>
>
> This performance gap occurs because **SSIM emphasizes structural similarity rather than precise pixel-level differences**. Many anomaly frames share similar structures with normal frames, causing SSIM to overlook subtle pixel differences. In contrast, MSE captures pixel-level differences more effectively, which aligns better with MLE’s objective of detecting small but meaningful differences between normal and abnormal frames.

---

> > ### Author Response · Authors · 2025-11-26
> >
> > Dear Reviewer BmTr,
> >
> > We would like to kindly check whether our responses have fully addressed your concerns. If you are satisfied with the revisions and feel that our changes have strengthened the submission in line with your suggestions, we would greatly appreciate an update to the review scores.
> >
> > If you require any additional clarification, please let us know—we would be happy to provide further details.
> >
> > Wishing you a happy Thanksgiving!

---

### Official Review · Reviewer_eCBT · 2025-10-31

**Soundness:** 3
**Presentation:** 3
**Contribution:** 3
**Rating:** 4
**Confidence:** 4

**Summary:**

The paper tackles single-scene, fully unsupervised video anomaly detection (VAD): training and detection are done directly on raw videos that contain a mix of normal and abnormal frames, with no labels at any stage. A convolutional autoencoder is trained with a dual loss: standard MSE reconstruction plus a Minimal Latent Entropy (MLE) regularizer (Rényi-2 entropy estimated via KDE). MLE encourages latent embeddings to concentrate around highdensity (normal) regions, effectively collapsing anomalous latents toward the normal cluster and widening the reconstruction gap between normal and abnormal inputs. Reconstruction quality is measured with the Pearson Correlation
Coefficient (PCC); anomalies are detected via a simple lower-tail threshold. On Donkeycar, Corridor, and UBnormal, the method reports high AUC and outperforms fully unsupervised baselines.

**Strengths:**

1. Simplicity with impact. Adding a single entropy regularizer (MLE) to a vanilla AE reliably amplifies the reconstruction gap, making anomalies easier to separate without complicated inference machinery.

2. Mechanistic clarity. The Rényi-2 + KDE formulation (with the pairwise Gaussian kernel view) and the kernel-gradient interpretation provide a convincing mechanism for why latent entropy minimization draws sparse anomaly embeddings toward the dominant normal manifold

**Weaknesses:**

1. Dependence on the “normal-majority” prior. The approach implicitly assumes normal frames dominate. Robustness under higher anomaly ratios (e.g., 50%) still needs fuller characterization despite some experiments.

2. Limited scope beyond single scene. The study targets single-scene deployments. This remains to be shown how well the trained model generalizes across multiple scenes/cameras without adaptation.

**Questions:**

1. When the majority prior breaks. If anomalies make up 30–60% of a video, how do the PCC distribution and the variance-aware threshold behave? Can the authors provide AUC curves versus anomaly ratio?

2. Beyond a single camera. What is the roadmap for transferring an AE+MLE trained on one scene to new cameras/scenes. Through camera-aware priors or domain adaptation, while preserving the same simple decision rule?

---

> ### Author Response · Authors · 2025-11-22
> **Response for Reviewer eCBT**
>
> We appreciate the reviewer’s insightful and detailed comments! We address each question along with additional experimental results.
>
> ### Weakness 1: "The approach implicitly assumes normal frames dominate. Robustness under higher anomaly ratios (e.g., 50%) still needs fuller characterization despite some experiments."
>
> Thanks for pointing out this important and interesting observation.
> While our method does need the assumption that normal content dominates, sometimes it remains effective when the anomaly ratio is high (e.g., 50%) for two reasons:
> * **Normal frames form a large, coherent latent cluster.** Normal scenes tend to be consistent across time and across videos, so their embeddings naturally concentrate into a single, dense cluster rather than dispersing.
> * **Anomalies are heterogeneous and form many small, sparse clusters.** Even within the same anomaly type (e.g., “protest”), the visual patterns vary significantly with different numbers of people, motions, and contexts. The large visual variation makes their abnormal latents remain scattered, preventing a dominant anomaly cluster.
>
> In summary, the normal and abnormal *content distribution* matters more than the *frame count*.
> A 50% normal–50% abnormal frame split does not translate to a 50%–50% split in latent-space density;
> It is closer to a 50% dense cluster (normal) vs. multiple small fragments (anomalies). This embedding asymmetry helps the normal cluster remain dominant.
>
> However, we still want to emphasize that such high anomaly ratios are **uncommon in real-world** camera systems. Analyzing too-high anomaly ratio is out of our scope.
> when the anomaly ratio is extremely high, performance becomes unstable. For example, at a 60% anomaly ratio on the Corridor and DonkeyCar datasets, the AUC drops to around 0.7. This occurs because there is *no noticeable visual variation* among these anomalies. Consequently, their distribution in the latent space is difficult to be disturbed from that of normal samples, making their reconstruction reach a level comparable to that of true normal samples. Nonetheless, our model still performs well when the ratio is below 50%.
>
> ### Weakness 2: "Limited scope beyond single scene." and Question 2: "What is the roadmap for transferring an AE+MLE trained on one scene to new cameras/scenes. Through camera-aware priors or domain adaptation, while preserving the same simple decision rule?"
> Thanks for your helpful suggestions. Multi-scene generalization is an important direction for our future work, but it is out of the scope of this paper. Our current study intentionally focuses on the single-scene setting because anomalies are often location-dependent: what is abnormal in one scene may be normal in another.
>
> Our unsupervised setting aligns with the single-scene setting. We simply collect raw video, train the model for this scene, and run real-time detection without any human effort. This “one camera–one model” pipeline remains practical and effective for many single-scene deployments.
>
> Moreover, we still conducted generalization transfer experiments using a single camera but with different anomalies. We trained a model on an unsupervised dataset containing one anomaly type (e.g., "normal + protest") and applied it to other anomaly types (e.g., "chasing," "fighting," and "suspicious object"). Our experiment shows that other unseen anomalies can also be detected with a noticeable reconstruction gap between the normal and abnormal frames, as shown in Figure 8 in the updated manuscript.
>
>
> ### Question 1: "If anomalies make up 30–60% of a video, how do the PCC distribution and the variance-aware threshold behave? Can the authors provide AUC curves versus anomaly ratio?"
> Thanks for this helpful suggestion, which indeed makes our experiments more complete.
> This week, we have run experiments with anomaly ratios from 20% - 60% across all datasets. The updated results are shown in the table below, and the PCC can be found in Figure 7 in the manuscript.
>
> When the anomaly ratio is low (e.g., below 30%), the AUC remains high (above 0.9). The reconstruction gap between normal and abnormal frames is clear. However, when the anomaly ratio exceeds 60%, the performance becomes unstable. For example, the AUC of the DonkeyCar dataset drops to around 0.6, while on the Corridor dataset it remains closer to 0.8. This behavior is **consistent with our assumption** that the method relies on normal content dominating the distribution.
>
> |  | 20%|30%|40%|50%|60%|
> | - |:-:| :-:|:-:|:-:|:-:|
> | Donkeycar | 1.000 | 0.811 | 0.740 | 0.657 | 0.691 |
> | Corridor  | 1.000 | 0.917 | 0.929 | 0.822 | 0.736 |
> | UBnormal | 1.000 | 1.000 | 1.000 | 1.000  | 0.900|
>
> At the same time, ratios above 50% are unusual in real-world surveillance scenarios, where anomalies are typically rare. Although this case is outside the primary scope, we appreciate the reviewer’s point and have added a detailed discussion and the requested PCC figures in the revision.

---

> ### Author Response · Authors · 2025-11-26
>
> Dear Reviewer eCBT,
>
> We would like to kindly check whether our responses have fully addressed your concerns. If you are satisfied with the revisions and feel that our changes have strengthened the submission in line with your suggestions, we would greatly appreciate an update to the review scores.
>
> If you require any additional clarification, please let us know—we would be happy to provide further details.
>
> Wishing you a happy Thanksgiving!

---

### Official Review · Reviewer_fBQu · 2025-11-01

**Soundness:** 2
**Presentation:** 3
**Contribution:** 2
**Rating:** 4
**Confidence:** 5

**Summary:**

This paper proposes an entropy-guided autoencoder for unsupervised video anomaly detection. By combining the reconstruction loss with a minimal latent entropy loss, the proposed dual-loss mechanism distinguishes normal and abnormal frames through reconstruction gap in reconstruction errors, thereby enabling effective anomaly detection. While the idea is conceptually reasonable, the overall contribution and novelty appear to be incremental.

**Strengths:**

1. The proposed minimum latent entropy loss is conceptually grounded in information theory and is derived from Rényi entropy with kernel density estimation, providing a clear and interpretable formulation for latent-space regularization.

2. The method reports high AUC scores on three datasets, showing consistent improvements over several baseline approaches, including semi-supervised and fully unsupervised methods.

**Weaknesses:**

1. The proposed method offers limited novelty, as its main contribution lies in combining reconstruction loss with a minimum latent entropy loss.

2. The compared methods appear somewhat outdated. In particular, the fully unsupervised approaches listed in Table 1 are relatively old, with the most recent baselines dating back to 2022, and the evaluation metrics used for comparison are also rather limited.

3. The experiments are conducted only on three relatively small datasets. Evaluating on more challenging and diverse datasets could better demonstrate the robustness and generalization capability of the proposed MLE-UVAD.

**Questions:**

1. How does the MLE loss prevent all embeddings, including those of normal samples, from collapsing into a trivial cluster? Is there any mechanism designed to preserve intra-class variability among normal samples?

2. In Equation (4), the value of α is empirically set to 2. Is there any theoretical rationale supporting this choice? Additionally, have the authors investigated how varying α affects model performance?

3. The authors mention that, as normal frames dominate the raw video, sparse anomalous embeddings tend to be pulled into the normal cluster. How would the proposed approach handle scenarios where the ratio of normal to abnormal frames is balanced, or where anomalous frames occur more frequently?

---

> ### Author Response · Authors · 2025-11-20
> **Follow-up questions: more datasets and baselines**
>
> As we prepare our comprehensive response, we wanted to reach out with a couple of questions below. First, thank you for your review and suggestions.
>
> ### Weakness 2: "The compared methods appear somewhat outdated, with the most recent unsupervised baselines back to 2022".
> We agree that including more recent methods is very important. We spent an additional week surveying newer fully unsupervised VAD techniques and added two competitive baselines:
> * **FRD-VAD** — 2024 Feature Reconstruction With Disruption for Unsupervised Video Anomaly Detection (Tao et al.)
> * **FUN-AD** — 2024 FUN-AD: Fully Unsupervised Learning for Anomaly Detection with Noisy Training Data (Im et al.)
>
> Both directly address unsupervised anomaly detection with contaminated training data.
> Our method still outperforms them on all datasets among different datasets. Soon, updated AUC results will be provided in the revised manuscript and a unified response to all reviews.
>
> Do you have any suggestions for other fully unsupervised, single-scene VAD baseline methods?
>
> ### Weakness 3: "The experiments are conducted only on three relatively small datasets. Evaluating on more challenging and diverse datasets could better demonstrate the robustness."
> We agree that using more datasets is beneficial. However, our fully unsupervised single-scene VAD setting imposes strict requirements on a dataset: one continuous video from a single camera/scene, mixed normal/abnormal frames, and frame-level ground truth labels in the video (only for evaluation).
>
> Most existing VAD datasets do not satisfy these constraints.
> Our chosen three datasets span driving, indoor surveillance, and synthetic anomaly scenes, covering both real-world and controlled-environment settings.
>
> Do you have any suggestions for an additional dataset? We would be happy to evaluate our model on it.

---

> ### Author Response · Authors · 2025-11-22
> **Response for Reviewer fBQu**
>
> Thank you for your careful and constructive evaluations! Below, we address each concern and include additional experiments in response to your feedback.
> ### Weakness 1: "The proposed method offers limited novelty; the main contribution lies in combining reconstruction loss with a minimum latent entropy loss."
> Sorry for not clarifying our novelty clearly. To address this concern, we would like to clarify it with two points:
> * **Conceptual novelty**: Our work is the first to introduce a **distribution-level latent entropy minimization mechanism** into the fully unsupervised, single-scene VAD setting. Prior methods depend on either normal-only data, pseudo-labeling, or supervised information.
> * **Practical significance**: Despite its simplicity, our method achieves **SOTA fully unsupervised performance** and matches recent semi-supervised methods. Our method can also be extended to widespread applications, especially under data contamination.
>
> Following your suggestions, we will explore more sophisticated architectures (e.g., prediction or transformer-based models) for future unsupervised VAD work.
> ### Weakness 2: "The compared methods appear somewhat outdated".
> Following our previous comments, we tested two 2024 unsupervised methods: FRD-VAD and FUN-AD. Our method still outperforms these two, achieving consistently higher AUC across three datasets. The AUC results are shown below.
> ||Donkeycar|Corridor|UBnormal|
> |-|:-:| :-:|:-:|
> | FRD-VAD| 0.751|0.811|0.935|
> | FUN-AD|0.633|0.478|0.499|
> | Ours|1.000|1.000|1.000|
> ### Question 1: "How does the MLE loss prevent all embeddings from collapsing into a trivial cluster? Is there any mechanism designed to preserve intra-class variability among normal samples?"
> Thank you for this excellent and insightful question! We fully appreciate it, and we address it with two points below:
>
> * **The reconstruction loss anchors normal embeddings** because accurate reconstruction requires the autoencoder to preserve its latent structure. If the entire latent space of normal samples collapses by MLE loss, reconstruction error increases, which is penalized. This naturally maintains the normal samples’ variability.
> * Sparse abnormal embeddings are **much farther** from the normal manifold. The normal embeddings are closer to each other. Because of this larger distance, entropy minimization pulls these sparse outliers toward the dense normal region much more strongly than it affects normal samples.
>
> **The two losses work jointly**:
> MSE prevents full collapse, while MLE selectively collapses sparse abnormal embeddings. Together, they produce a stable latent space with preserved normal variation and collapsed anomalies.
> ### Question 2: "In Equation (4), the value of α is empirically set to 2. Is there any theoretical rationale supporting this choice?"
> We appreciate the reviewer for catching this detail! The choice of α = 2 is intentional and follows common practice in information-theoretic learning:
> 1. **Closed-form, differentiable estimator:** α = 2 yields a closed-form expression for the Rényi entropy estimator when using Gaussian kernels. This makes the optimization process stable and efficient.
> For α ≠ 2, no such closed-form exists, and gradient computation becomes significantly more difficult and less reliable.
> 2. **Consistency with prior literature:** Many related works set α = 2 for the same practical reason, including:
> - *Chen, Badong, et al. "Minimum error entropy Kalman filter." IEEE Transactions on Systems, Man, and Cybernetics: Systems, 2019.*
> - *Feng, Zhenyu, et al. "Distributed minimum error entropy Kalman filter." Information Fusion, 2023.*
> - *Dang, Lujuan, et al. "Cubature Kalman filter under minimum error entropy with fiducial points for INS/GPS integration." IEEE/CAA Journal of Automatica Sinica, 2021.*
> - *Luo, Yarong, et al. "A novel perspective of the Kalman filter from the Rényi entropy." Entropy, 2020.*
>
> We added these citations and clarified the rationale in the main text.
> ### Question 3: "How would the proposed approach handle scenarios where the ratio of normal to abnormal frames is balanced, or where anomalous frames occur more frequently?"
> This is an important question! We answered this question by performing additional experiments with anomaly ratios from 10% to 60% across all datasets (See Fig.7 and Table 4 in PDF). Performance is stable up to ~40% anomalies with relatively high AUC around 0.9.
>
> However, when anomalies **exceed 50%**, performance drops noticeably with a low AUC around 0.7. The MLE term loses guidance because the latent space is no longer dominated by normal patterns. As the overall entropy is reduced, both normal and abnormal embeddings collapse together, resulting in poor reconstruction for both. Therefore, reconstruction quality cannot be used to reliably detect anomalies. This is **consistent with our core assumption** that our method works when anomalies are naturally sparse, which aligns with most real-world video surveillance scenarios.

---

> ### Author Response · Authors · 2025-11-26
>
> Dear Reviewer fBQu,
>
> We would like to kindly check whether our responses have fully addressed your concerns. If you are satisfied with the revisions and feel that our changes have strengthened the submission in line with your suggestions, we would greatly appreciate an update to the review scores.
>
> If any additional clarification is needed, please let us know—we would be happy to provide further details.
>
> Wishing you a Happy Thanksgiving!

---

### Meta-Review · Area_Chair_LUXZ · 2026-01-07

**Summary:**

The initial ratings are 4,4,6,4. This paper proposes a fully unsupervised VAD. It uses a convolutional autoencoder and introduces a novel MLE loss in combination with reconstruction loss to address the problem of detecting anomalies in a single-scene video setting. The proposed system aims to detect abnormal events in videos without any labeled data, making it suitable for privacy-sensitive applications.Experiments show the effectiveness of this method.

Strengths:
(1)The proposed minimum latent entropy loss is conceptually grounded in information theory and is derived from Rényi entropy with kernel density estimation, providing a clear and interpretable formulation for latent-space regularization.
(2)The method is evaluated on the diverse datasets, outperforming both fully supervised and unsupervised baseline VAD methods.

Weaknesses:
(1)The proposed method offers limited novelty, as its main contribution lies in combining reconstruction loss with a minimum latent entropy loss.
(2)The paper uses the same unlabelled data for both training and testing, which leads to potential information leakage. It also tracks AUC with true labels, which can bias results.

**Reviewer Concerns:**

Most concerns of Reviewer BmTr and fBQu were addressed by the rebuttal, and Some main concerns of  Reviewer eCBT and HpRQ  are still outstanding.

**Reviewer Scores:**

None.

---

### Decision · Program_Chairs · 2026-01-26

Reject